# Incorporation of volcanic $SO_2$ emissions in the Hemispheric CMAQ (H-CMAQ) version 5.2 modeling system and assessing their impacts on sulfate aerosol over Northern Hemisphere

Syuichi Itahashi[1], Rohit Mathur[2], Christian Hogrefe[2], Sergey L. Napelenok[2], and Yang Zhang[3]

[1] Environmental Science Research Laboratory, Central Research Institute of Electric Power Industry (CRIEPI), 1646 Abiko, Abiko, Chiba 270–1194, Japan
[2] Center for Environmental Measurement and Modeling, Office of Research and Development, U.S. Environmental Protection Agency, Research Triangle Park, NC 27711, U.S.A.
[3] Department of Civil and Environmental Engineering, Northeastern University, Boston, MA 02115, U.S.A.

*Correspondence to*: Syuichi Itahashi (isyuichi@criepi.denken.or.jp)

**Abstract**

The state-of-the-science Community Multiscale Air Quality (CMAQ) Modeling System has recently been extended for hemispheric-scale modeling applications (referred to as H-CMAQ). In this study, satellite-constrained estimation of the degassing $SO_2$ emissions from 50 volcanos over the northern hemisphere is incorporated into H-CMAQ, and their impact on tropospheric sulfate aerosol ($SO_4^{2-}$) levels is assessed for 2010. The volcanic degassing improves predictions of observations from the Acid Deposition Monitoring Network in East Asia (EANET), the United States Clean Air Status and Trends Network (CASTNET), and the United States Integrated Monitoring of Protected Visual Environments (IMPROVE). Over Asia, the increased $SO_4^{2-}$ concentrations were seen to correspond to the locations of volcanoes, especially over Japan and Indonesia. Over the U.S.A., the largest impacts occurred over the central Pacific caused by including the Hawaiian Kilauea volcano while the impacts on the continental U.S.A. were limited to the western portion during summertime. The emissions of the Soufriere Hills volcano located on Montserrat Island in the Caribbean Ocean affected the southeastern U.S.A. during the winter season. The analysis at specific sites in Hawaii and Florida also confirmed improvements in regional performance for modeled $SO_4^{2-}$ by including volcanoes $SO_2$ emissions. At the edge of the western U.S.A., monthly-averaged $SO_4^{2-}$ enhancements greater than 0.1 μg/m$^3$ were noted within the boundary layer (defined as surface to 750 hPa) during June-September. Investigating the change on $SO_4^{2-}$ concentration throughout the free troposphere revealed that although the considered volcanic $SO_2$ emissions occurred at or below the middle of free troposphere (500 hPa), compared to the simulation without the volcanic source, $SO_4^{2-}$ enhancements of more than 10% were detected up to the top of the free troposphere (250 hPa). Our model simulations and comparisons with measurements across the Northern Hemisphere indicate that the degassing volcanic $SO_2$ emissions are an important source and should be considered in air quality model simulations assessing background $SO_4^{2-}$ levels and their source attribution.

# 1 Introduction

Airborne sulfate ($SO_4^{2-}$) is one of the major components of tropospheric particulate matter worldwide (Zhang et al., 2007) and plays important roles in modulating the earth-atmosphere energy budget, atmospheric circulation, cloud properties, and precipitation (Seinfeld and Pandis, 2016). $SO_4^{2-}$ is produced via the aqueous- and gas-phase oxidation of sulfur dioxide ($SO_2$), and these processes are well understood (Seinfeld and Pandis, 2016). The dominant sources of $SO_2$ emissions are attributed to anthropogenic activity (Warneck and Williams, 2012). The global anthropogenic $SO_2$ emissions peaked in the early 1970s with around 130 Tg/yr and then decreased; however, this emission trend has contrasting characteristics over the U.S.A. and Asia (e.g., Smith et al., 2011; Xing et al., 2015a). Anthropogenic $SO_2$ emissions from the U.S.A. showed a peak in the early 1970s with 30 Tg/yr and subsequently decreased (Smith et al., 2011). Publicly available observational records have begun from the late 1980s over the U.S.A., and it has been confirmed that $SO_4^{2-}$ concentration in the U.S.A. decreased during the early 1990s through 2010 in response to these reductions in $SO_2$ emissions as evidenced in analyses of observational aerosol composition (e.g., Hand et al., 2012; Gan et al., 2015). On the other hand, anthropogenic $SO_2$ emissions across Asia, especially China, have shown a continuous increase since 1970 (Smith et al., 2011) up to 2006 and then decreased (Li et al., 2017) in response to control measures. These multi-decadal changes in $SO_2$ emissions have resulted in not only contrasting changes in tropospheric $SO_4^{2-}$ levels but also in aerosol radiative effects (e.g., Wild et al., 2009; Xing et al., 2015b), their feedback on atmospheric dynamics and air quality (e.g., Xing et al., 2016), and acid deposition (e.g., Zhang et al., 2018; Mathur et al., 2020).

As $SO_2$ emission control is reducing regional airborne $SO_4^{2-}$, quantifying the relative contribution of long-range transported $SO_4^{2-}$ and the portion attributable to natural sources is becoming increasingly important. For instance, regional haze assessments require quantification of visibility impairment that are associated with anthropogenic enhancements over natural visibility levels, which in turn necessitates accurate quantification of the contribution of natural sources. Next to the anthropogenic emissions, volcanic emissions have an important contribution to $SO_2$ emissions (Warneck and Williams, 2012). A time-averaged inventory of volcanic emissions was estimated as 13 Tg-$SO_2$/yr during early 1970s to 1997, and these volcanoes are mostly located in the Pacific Rim region (Andres and Kasgnoc, 1998). Volcanic emission fluxes can be measured in several ways such as a correlation spectrometer (COSPEC), but observable volcanic eruptions are limited in time and location. Satellite observations are now proving to be a useful approach to monitor the volcanic emissions and have provided a global volcanic $SO_2$ emission inventory since 1978 (Carn et al., 2016). Accurate representation of volcanic emissions requires quantification of emissions emitted from both persistent degassing and sporadic eruptions. Recently, a decadal-scale global volcanic $SO_2$ emissions inventory for the 2005-2015 period, constrained by the Ozone Monitoring Instrument (OMI) has been established (Carn et al., 2017). The enhanced methodology with greater sensitivity allows the detection of emissions as low as approximately 6 Gg-$SO_2$/yr for low-altitude volcanoes, and covers a total of 91 persistently degassing volcanic $SO_2$ sources. The volcanic $SO_2$ emissions from degassing are relatively stable at 23.0±2.3 Tg-$SO_2$/yr during 2005-2015, and the highest amount was approximately 26 Tg-$SO_2$/yr in 2010. In terms of the volcanic activity, sporadic eruptions inject into the

atmosphere comparatively large amounts of $SO_2$ emissions. For example, one of the largest eruptions in the 1990s was that of Mt. Pinatubo in June 1991 which was estimated to have injected 18 Tg of $SO_2$ emissions into the atmosphere (Smithsonian Institution, 2020). The injection heights of that eruption reached to more than 30 km and caused the largest perturbations ever observed in the chemical state of the stratosphere and the earth-atmosphere radiation budget (McCormick, et al., 1995).

However, such eruptive emissions are temporary. According to the comparison between the degassing and eruptive emissions, Carn et al. (2017) estimated that during 2005-2015, volcanic activity contributed about 23 Tg/yr of $SO_2$ due from degassing while eruptive $SO_2$ emissions ranged from 0.2 to 10 Tg/yr of $SO_2$. Therefore, understanding the behavior of the degassing $SO_2$ emissions and its contributions to airborne $SO_4^{2-}$ levels is important. For example, it was reported that although volcanic $SO_2$ emissions contributed 15% to the total sulfur emissions, it attributed 27% of the tropospheric sulfur budget (Lamotte et al.,

2021).

        Numerical modeling is a useful tool to characterize source-receptor relationships. Regional modeling studies have already indicated that volcanic $SO_2$ emissions are one of the main sources of $SO_4^{2-}$ over Japan (Itahashi et al., 2017a; 2017b; Itahashi, 2018; Itahashi et al., 2019). $SO_2$ emissions from volcanos in the Pacific Rim region not only regulate tropospheric $SO_4^{2-}$ levels in surrounding countries but through long-range transport can also potentially impact $SO_4^{2-}$ distributions over the

Pacific and background levels in the western U.S.A. Liu et al. (2008) for instance suggest that west to east across the North Pacific, sulfate originating from East Asia sources contributed approximately 80%–20% of sulfate at the surface, but at least 50% at 500 hPa. Taking into consideration that volcanoes are mainly over the Pacific Rim area and the seasonally prevalent cross-Pacific transport patterns, volcanic $SO_2$ emissions could also affect $SO_4^{2-}$ concentrations over North America. While previous studies have attempted to quantify global tropospheric $SO_4^{2-}$ budgets (e.g., Chin et al., 1996; Chin and Jacob, 1996),

the assessments are representative of conditions in the late 1980's to early 1990's. Since anthropogenic $SO_2$ emissions have changed significantly over the past several decades, and since recent studies provide improved constraints of volcanic $SO_2$ emissions, the work summarized in this manuscript attempts to assess the contributions of volcanic $SO_2$ emissions on tropospheric $SO_4^{2-}$ distributions across the Northern Hemisphere and North America. We specifically focus on assessing the impacts of the persistent degassing volcanic $SO_2$ emissions.

This manuscript is organized as follows. In section 2, the modeling system and simulation set up are described along with the ground-based observations used to evaluate the model performance. In section 3, the model results and comparisons with observations are presented and the impact of including volcanic $SO_2$ emissions is discussed. Finally, section 4 summarizes the key results and limitations of this work, and discusses future perspectives.

## 2 Methodology

### 2.1 Hemispheric CMAQ modeling system and its set up

The Community Multiscale Air Quality (CMAQ) modeling system version 5.2 extended for hemispheric applications (H-CMAQ) (Sarwar et al., 2015; Xing et al., 2015a; Mathur et al., 2017; Itahashi et al. 2020a, 2020b) is used to incorporate and assess the impacts from volcanic $SO_2$ emissions. H-CMAQ is configured to cover the entire Northern Hemisphere, utilizing a polar stereographic horizontal discretization of 187×187 grid points with a grid spacing of 108 km and a terrain-following vertical coordinate system with 44 layers of variable thickness from the surface up to 50 hPa. The emission datasets for the base case H-CMAQ simulation are based on the Hemispheric Transport of Air Pollution version 2 (HTAP2) experiments. The details were described in previous studies (Janssens-Maenhout et al., 2015; Pouliot et al., 2015; Galmarini et al., 2017; Hogrefe et al., 2018). Over the modeling domain, a total of 105.8 Tg/yr of $SO_2$ emissions are emitted. The spatial distribution of $SO_2$ emissions is shown in Fig. 1a with high $SO_2$ emissions from fossil fuel combustion activities in North America, Europe including western Russia, and Asia, with relatively higher amounts across regions in China and India as described in Janssens-Maenhout et al. (2015). The CMAQ configuration employed the CB05 gas-phase chemical mechanism and the aero6 module with nonvolatile primary organic aerosol (POA) (Appel et al., 2017). CMAQ treats one gas-phase oxidation and five aqueous-phase oxidations for converting S(IV) (i.e., sulfur compounds with oxidation state 4) into S(VI) (i.e., sulfur compounds with oxidation state 6). The gas-phase oxidation involves reaction with hydroxyl (OH) radical, and five aqueous-phase reactions involve oxidation by hydrogen peroxide ($H_2O_2$), ozone ($O_3$), oxygen ($O_2$) via Fe and Mn catalysis, methyl hydrogen peroxide (MHP), and peroxyacetic acid (PAA). The meteorological fields to drive H-CMAQ are derived from simulations with the Weather Research and Forecasting (WRF) model version 3.6.1. The WRF model is configured to use the rapid radiative transfer model for global climate models (RRTMG) radiation scheme for both longwave and shortwave (Iacono et al., 2008), Morrison double-moment scheme (Morrison et al., 2009) and the Kain-Fritsch (KF) cumulus parameterization (Kain, 2004) for microphysics and cumulus parameterization, and Mellor-Yamada-Janjic scheme for planetary boundary layer (Janjic et al., 1994). In this study, the entire year of 2010 was simulated to analyze the volcanic emission impacts over the Northern Hemisphere, as the $SO_2$ emissions from degassing during the 2005-2015 period were highest in 2010. Also note that emission estimates presented in Carn et al., 92017) suggest that degassing dominate over eruptive $SO_2$ emissions during the same time period. The WRF simulations used nudging for wind, temperature, and water vapor fields towards NCEP final analysis (FNL) of 1° spatial and 6 h temporal resolution (NCEP, 2020) over the entire vertical model extent. WRF simulation started from 1 January 2009 to set one year spin-up time prior to the analysis period the year of 2010 as recommended by Mathur et al. (2017). The CMAQ simulation were initialized on 1 December 2009 with three-dimensional chemical fields from prior model simulations for 2010 by Hogrefe et al. (2018).

Earlier development and applications with the H-CMAQ modeling system have not considered volcanic $SO_2$ emissions. In this study, degassing volcanic $SO_2$ emissions as annual amount estimated by Carn et al. (2017) are incorporated into H-CMAQ. The estimated emissions of $SO_2$ from the 50 volcanos within our northern hemisphere modeling domain (Figure

1b) is 12.7 Tg/yr. Considering the characteristics of degassing process from volcanoes and the lack of any other information to accurately specify its temporal variations, we use a constant $SO_2$ emission rate in H-CMAQ based on the annual $SO_2$ emission estimates by Carn et al. (2017). Table 1 provides detailed information on these 50 volcanoes including name, location (longitude and latitude), altitude (as above sea level (a.s.l.)), and $SO_2$ emission rate. Kilauea located in Hawaii is estimated to

have the highest amount of degassing $SO_2$ emissions. Taking into accounts the characteristics of the degassing process, volcanic degassing $SO_2$ emissions were allocated into the model vertical layer corresponding to the volcano's altitude. This approach was also taken in another global model analysis (Chen et al., 1996; Ge et al., 2016). Even though in principle volcanic degassing emissions could be assigned to the first model layer because CMAQ uses a terrain following vertical coordinate system, the 108 km grid spacing used in CMAQ does not allow the model to adequately resolve localized terrain peaks such

as volcanoes and assigning their emissions to the first model layer would not account for the fact that in reality these emissions typically occur above the mixed layer. Therefore, the vertical layer to which volcanic degassing emissions were assigned was determined by first calculating the difference between the altitude of a given volcano (Table 1) and the CMAQ terrain height for the cell in which it is located, and then determining the vertical CMAQ layer corresponding to this difference. A schematic of the resultant assigned vertical layer is illustrated in Fig. 2. Most of volcanoes are located between altitudes of 500-5000 m

above sea level (a.s.l) (Table 1), and these correspond to layers 11-26 in the current model configuration.

**2.2 Ground-based Observations**

As seen from Fig. 1 (b), the majority of the degassing volcanoes are located in the Pacific Rim region and their impacts on $SO_4^{2-}$ levels in Japan have previously been studied (Itahashi et al., 2017a, b, 2019; Itahashi, 2018). To assess model

performance in the Asian region, ground-based surface $SO_4^{2-}$ observations were obtained from the Acid Deposition Monitoring Network in East Asia (EANET) program (EANET, 2020). During 2010, filter-pack measurements of $SO_4^{2-}$ are available at 35 EANET sites with the following geographic distribution: 1 site in China, 3 sites in the South Korea, 12 sites in Japan, 2 sites in Mongolia, 4 sites in Russia, 5 sites in Thailand, 2 sites in Vietnam, 2 sites in Philippines, 3 sites in Malaysia, and 1 site in Indonesia. Because the four-stage filter pack method in EANET measured total suspended particle, non-seasalt (nss)-$SO_4^{2-}$

was calculated as nss-$SO_4^{2-}$ = $SO_4^{2-}$ − 0.251×$Na^+$ in order to exclude the coarse-mode $SO_4^{2-}$ produced with seasalt. For the simplicity, nss-$SO_4^{2-}$ for EANET observation is written as $SO_4^{2-}$ through this manuscript. Most of sites report two-week measurements, but some sites provide weekly or daily measurements. All available observation data were used in this study. As we have discussed in previous work (Itahashi et al., 2020b), Asian air pollution can impact on air quality over the U.S.A.; however, the magnitude of volcanic emissions on tropospheric $SO_4^{2-}$ levels over North America has not been well studied. To

evaluate the impact of volcanic $SO_2$ emissions located in the northern hemisphere on model performance, surface observations over the U.S.A. were obtained from the Clean Air Status and Trends Network (CASTNET) which covered remote and rural sites mostly over eastern U.S.A. (CASTNET, 2020) and the Integrated Monitoring of Protected Visual Environments (IMPROVE) which covered remote sites mostly over western U.S.A. (IMPROVE, 2020). Sampling frequency is weekly and

daily (1-in-3 day) for CASTNET and IMPROVE, respectively. A total of 84 CASTNET sites and 170 IMPROVE sites were available in 2010. Because of the coarse resolution of H-CMAQ simulations, measurements at urban sites were not considered in this study. To further examine the model's ability to represent the tropospheric sulfur distributions and budget, ambient $SO_2$ concentration at CASTNET sites and vertically integrated $SO_2$ column concentration measured by the OMI sensor were also analysed. The OMI measured $SO_2$ column was taken from the level 3 daily global sulfur dioxide product (OMSO2e) gridded into 0.25° (Krotkov et al., 2015). This gridded data were mapped to the H-CMAQ modeling domain and grid structure. This product contains the total column of $SO_2$ in the planetary boundary layer (PBL) with its center of mass altitude (CMA) at 0.9 km, and lacked in the vertical sensitivity. Our previous study indicated that 1 km depth for CMA in the modeled $SO_2$ column yielded best comparisons with satellite-measured $SO_2$ column over East Asia (Itahashi et al., 2017a). The observed deficit grids by satellite were also considered in the analysis of modeled $SO_2$ column. The same approach to compare and evaluate $SO_2$ column was used in this study. Furthermore, $SO_4^{2-}$ concentration in precipitation, precipitation amount, and $SO_4^{2-}$ wet deposition were also evaluated relative to observations at EANET sites over Asia and the National Atmospheric Deposition Program's National Trends Network (NADP/NTN) over the U.S.A. (NADP, 2021). Measurements by wet-only sampler are available at 49 EANET sites with daily interval at most sites and weekly or 10-days interval at others sites (EANET, 2020). Observation of volume-weighted concentration in precipitation and total weekly wet deposition are available at 240 NADP sites.

To evaluate model performance with ground-based observations, the Pearson's correlation coefficient (R) with student's $t$-test is used for assessing the statistical significance level. The normalized mean bias (NMB) and the normalized mean error (NME) are also calculated as follows (e.g., Zhang et al., 2006; Itahashi et al., 2020a).

$$R = \frac{\sum_1^N(O_i - \bar{O})(M_i - \bar{M})}{\sqrt{\sum_1^N(O_i - \bar{O})^2}\sqrt{\sum_1^N(M_i - \bar{M})^2}} \qquad (1)$$

$$NMB = \frac{\sum_1^N(M_i - O_i)}{\sum_1^N O_i} \qquad (2)$$

$$NME = \frac{\sum_1^N |M_i - O_i|}{\sum_1^N O_i} \qquad (3)$$

where, N is the total observation number, $O_i$ and $M_i$ represent each individual observation and model result respectively, and $\bar{O}$ and $\bar{M}$ represent the arithmetical mean of observations and model results respectively. Based on a review of model performance over North America simulated by regional-scale air quality models, Emery et al. (2017) suggested threshold values of R > 0.70, NMB < ±10%, and NME < 35% as performance goal, and threshold values of R > 0.40, ±10% < NMB < ±30%, and 35% < NME < 50% as performance criteria for daily $SO_4^{2-}$.

## 3 Simulation Results and Discussion

### 3.1 Model Evaluation

To provide an overview of the modeling results, we first present the annual-average $SO_4^{2-}$ simulated by the base H-CMAQ configuration in Fig. 3 (a). High concentrations of $SO_4^{2-}$ (greater than 5 μg/m$^3$; red color in Fig. 3) are noted over East Asia, some parts of India, and the Arabian Peninsula corresponding to the intense $SO_2$ emissions shown in Fig. 1. The concentrations of $SO_4^{2-}$ over Europe and U.S.A. were mostly 0.5-2.5 μg/m$^3$ (blue color to green color in Fig. 3). The performance of this base case H-CMAQ simulation was evaluated over Asia and the U.S.A through comparison with $SO_4^{2-}$ measurements by EANET, CASTNET, and IMPROVE. The results of statistical analysis using R, NMB, and NME are listed in Table 2. Detailed maps of model results over Asia and the U.S.A. with overlaid distribution of surface observations are shown in Fig. 4. Fig. 4 also contains a scatter plot between base H-CMAQ and surface observations are also shown; data for each month is shown using a different color. Significant scatter is noted in the correlation between the modeled and observed concentrations with an R of 0.43 over Asia (Table 2). Recent analysis of 12 regional models participating in the MICS-Asia model intercomparison study (Chen et al., 2019) however also revealed moderate correlations (0.46-0.79) with EANET observation. In terms of NMB and NME, NMB was −37.6% and NME was 67.0% in this study (Table 2). MICS-Asia showed NMB of −19.1% as model ensemble mean, but ranged from −67.0% to +69.3% for individual regional models (Chen et al., 2019). The H-CMAQ base case performance statistics were comparable or slightly worse compared to the previous regional-scale modeling studies (e.g., Itahashi, 2018; Itahashi et al., 2018; Yamaji et al., 2020; Chatani et al., 2020). This in part results from the inability of the coarse model grid resolution to resolve localized high pollution episodes as seen in the scatter-plot in Fig. 4. Over the U.S.A., the spatial distribution patterns with low $SO_4^{2-}$ in the western U.S.A. and comparatively higher values in the eastern U.S.A. are well captured in the base H-CMAQ simulation, though the model tended to underestimate across sites in the eastern U.S.A. The observed annual averaged values by CASTNET were ranged between 2.5 and 3.4 μg/m$^3$ over eastern U.S.A. The scatter-plot also verified the reasonable correspondence between model and observations. A winter minimum and summer maximum is noted both in the modeled and observed $SO_4^{2-}$ values across the U.S.A. driven by expected variations in intensity of oxidant chemistry and conversion of S(IV) to S(VI). The statistical scores of R, NMB, and NME were within or close to the performance criteria proposed by Emery et al. (2017) over the U.S.A., and were also comparable to previous regional-scale modeling studies (e.g., Zhang et al., 2009; Zhang et al., 2013). Overall, despite the use of coarse horizontal grid resolution of 108 km in H-CMAQ, model performance statistics were generally within the model performance statistics noted for other the regional modeling applications. One possible contributor to the noted $SO_4^{2-}$ underestimation over Asian region in the base H-CMAQ could be from the missing of volcanic $SO_2$ emissions, especially in the Pacific Rim. Comparisons of model estimated $SO_2$ column distributions with satellite derived values are presented in Fig. S1 in the Supplement. The R value of 0.48 indicated the general agreement for the spatial distribution pattern of $SO_2$ column over the Northern Hemisphere with higher concentrations in regions with high anthropogenic $SO_2$ emissions (illustrated in Fig. 1); however, high $SO_2$ column over

Hawaii evident in the satellite retrieval was not captured in the base H-CMAQ simulation due to the lack of volcanic $SO_2$ emissions. The impact of introducing the degassing volcanic $SO_2$ emissions is further discussed in the following sections.

### 3.2 Impact of incorporating volcanic $SO_2$ emissions

The annual averaged $SO_4^{2-}$ simulated after incorporating degassing volcanic $SO_2$ emission in H-CMAQ is shown in Fig. 3 (b), and the increase in concentrations relative to the base H-CMAQ is shown in Fig. 3 (c) as absolute value and in Fig. 3 (d) as the percentage change from the original H-CMAQ simulation. Impacts of including volcanic $SO_2$ emissions on tropospheric $SO_4^{2-}$ are noted across the Pacific and up to the western coastline of the U.S.A. It should be noted that even though the degassing volcanic emissions are allocated to upper model layers (see, Fig. 2 and Table 1), they get transported through much of the troposphere with non-trivial impacts detected at the surface level. Increases of at least 0.1 μg/m$^3$ in $SO_4^{2-}$ concentration were simulated except over central Asia, equatorial and high latitude regions, and the Atlantic ocean, and most of the U.S.A. The maximum increase of greater than 1.0 μg/m$^3$ on an annual average basis was seen over the central Pacific (Fig. 3(c)). This increase is primarily attributed to $SO_2$ emission from Kilauea in Hawaii which is estimated to have the highest degassing emissions (see Table 1). In addition, a moderate increase of up to 1.0 μg/m$^3$ on an annual average basis was found over the Antilles islands in the Caribbean Sea. This was related to the volcanic activity of Soufriere Hills volcano located in Montserrat (No. 5 in strength; Table 1). Compared to the broad impacts found over Pacific Rim region, the impact of incorporating degassing volcanic $SO_2$ emission was limited over Europe and Africa. Increased concentration ranging between 0.1-0.3 μg/m$^3$ simulated over southern Europe and northern Africa region were caused by volcanoes in Italy (Nos. 8 and 29 in Table 1), Ethiopia (No. 45 in Table 1), and Yemen (No. 47 in Table 1). Because only four degassing volcanoes in this region are considered in this study (see, Fig. 1 (b)), the impact itself was lower compared to other regions. In the terms of the percentage changes relative to the original H-CMAQ (depicted in Fig. 3 (d)), increased concentration greater than 1.0 μg/m$^3$ seen in Fig. 3 (c) corresponded to +200% change, and that of 0.1 μg/m$^3$ corresponded to about a +10% change. Over North America and the polar region, the increased absolute concentration was less than 0.1 μg/m$^3$, whereas the percentage increase change was 10-30%. For the annual average, it was found that the degassing volcanic $SO_2$ emissions increased $SO_4^{2-}$ concentrations less than 0.1 μg/m$^3$ over the entire U.S.A. but this still represented a 10-20% increase over the western U.S.A.

The impacts on modeling performance by including volcanic $SO_2$ emissions are discussed based on the statistical analysis scores. In terms of the statistical analysis listed in Table 2, NMB and NME over Asia compared to EANET observation were improved, though the R values were not impacted significantly. This result was consistent with our previous findings that suggested volcanoes are an important source of tropospheric sulfur in East Asia (Itahashi et al., 2017; Itahashi, 2018; Chatani et al., 2021). Over the U.S.A., the base H-CMAQ had better modeling performance compared to Asia, and NMB and NME were improved when volcanic degassing emissions were included but R did not change. The improvements were noticeable in the comparison with observations at IMPROVE sites located in the western U.S.A., NMB showed close agreement between H-CMAQ and IMPROVE observation. Both simulations of original H-CMAQ and H-CMAQ incorporating degassing $SO_2$

emissions showed model overestimation for ambient $SO_2$ concentration when compared to CASTNET observation as listed in Table S1 in the Supplement. The model performance statistics were comparable for the case incorporating degassing volcanic $SO_2$ emissions; indicating that additional $SO_2$ emissions from volcano were fully oxidized to $SO_4^{2-}$ and thus have greater impact on changes in model performance for $SO_4^{2-}$. The overestimation of $SO_2$ could be attributed to the insufficient oxidation process from $SO_2$ to $SO_4^{2-}$ in the modeling system, because $SO_4^{2-}$ were still underestimated by incorporating volcanic $SO_2$ emissions. Coarse grid resolution leading to artificial dilution of $SO_2$ emissions, could also contribute to overestimation of predicted ambient $SO_2$ levels relative to measurements at remote locations. The detailed discussion using conversion rate is further presented in next Section 3.3. The comparison of $SO_2$ column with satellite observation showed the improvement of R value from 0.48 to 0.56 due to improvements in representation of the spatial variability in tropospheric $SO_2$ distributions resulting from capturing the high $SO_2$ column in vicinity of active volcanos such as Kilauea as illustrated in Fig. S1 in the Supplement. The evaluation for removal processes such as wet deposition is tabulated in Table S2 in the Supplement. The base H-CMAQ simulation tended to underestimate both $SO_4^{2-}$ concentration in precipitation and wet deposition. The inclusion of volcanic $SO_2$ emission sources led to slight improvements in the NMB and R, but the NME was largely unaltered. From these evaluations, it is concluded that the incorporation of degassing volcanic $SO_2$ emissions helps to improve performance of simulated $SO_4^{2-}$ in H-CMAQ by a small margin.

The impacts of degassing volcanic $SO_2$ emissions on seasonal $SO_4^{2-}$ distributions and long-range transport were further analyzed by examining monthly mean contributions as illustrated in Fig. 5. Similar to the annual mean distributions shown in Fig. 3 (c), the maximum impact was seen over the central Pacific associated with emissions from the Kilauea volcano throughout the year. Regarding its monthly variation, the increased concentration at the surface level during spring to summer season (from April to September) were higher than those during winter (especially, December and January). The higher impacts during summer result both from higher rates of $SO_2$ to $SO_4^{2-}$ conversion as well as enhanced convective mixing. In contrast during winter lower conversion rates and a more stably stratified atmosphere result in reduced $SO_4^{2-}$ from volcanic emissions at the surface. The enhanced $SO_4^{2-}$ associated with emissions from the Kilauea volcano (with highest emission rate as indicted in Table 1) stretch across the central Pacific to the western shores of the U.S., with enhancements in surface-level $SO_4^{2-}$ on a monthly-mean basis greater than 0.1 $\mu g/m^3$ across portions of the western U.S. The contribution of volcanoes located on the Kamchatca Peninsula (Nos. 2, 6, 12, 14, 16, 19 and others in Table 1) are estimated to be comparatively lower on an annual average basis, but may be higher episodically under conducive transport condition. During the winter season, the impacts from Kilauea did not reach to the western U.S.A., however, the impacts from Soufriere Hills located in Montserrat (No. 5 in Table 1) reached the southern U.S.A. Further analysis focused on specific sites is discussed in next section.

### 3.3 Impacts of volcanic $SO_2$ emissions at specific sites

Detailed analysis at an observation site located in Hawaii was further conducted to evaluate model performance. According to the observation summary (WRAP, 2020), a large fraction of the measured $SO_4^{2-}$ at this location is attributed to

SO$_2$ emissions associated with volcanic activity from Kilauea. A total of three IMPROVE sites are located in the state of Hawaii. One of site is located in the Hawaii Volcanoes National Park and is in vicinity of the Kilauea volcano (site name is HAVO1). Comparison of model and observed temporal variation in SO$_4^{2-}$ at this site are displayed in Fig. 6 (a). The observation showed higher concentration during the winter season, and the average concentration through 2010 was 2.37 μg/m$^3$. The base

H-CMAQ simulation showed largely invariant concentrations through the year with an annual average value of 0.87 μg/m$^3$ and a negative correlation against observation. The statistical scores of NMB showed high negative bias because the base H-CMAQ did not capture the higher concentration observed during winter. By including the degassing volcano emissions, the model showed spikes for higher concentration days. Though the observed maximum peak of 30 μg/m$^3$ on 3 January was not fully captured, the simulated values of around 10 μg/m$^3$ by incorporating the volcano emissions were significantly enhanced

relative to the base model. The result showed that the averaged concentration was 2.87 μg/m$^3$, and NMB showed +21.2% with moderate correlation by R of 0.36. The inclusion of degassing volcanoes led to better performance at this specific site; however, the deterioration in the value of NME should be considered. While the H-CMAQ simulations with incorporated volcanoes also showed lower SO$_4^{2-}$ concentrations during summer season compared to winter, the simulated summertime values were higher than observed and led to a persistent positive bias during this season. A potential reason for this behavior could be the treatment

of degassing volcanoes SO$_2$ emissions as constant flux in this study. For example, a case study targeted to quantify the air quality impacts of the Kilauea eruption in 2018 showed that the importance of temporal variation of emissions and plume rise calculation (Tang et al., 2020). Our results indicate general improvements of model performance by including degassing volcano emissions represented by a constant temporal and vertical profile, but refining the treatment of these aspects should be studied in future work. In terms of SO$_2$ concentration, IMPROVE sites do not measure it. Based on an intensive

observational study at Kilauea during January-February 2013, SO$_2$ concentrations showed large variations from below 1 ppbv under conditions of no influence of the volcanic plume to over 3,000 ppbv when airmasses influenced by the volcanic plume were sampled (Kroll et al., 2015). In the base H-CMAQ calculation, the daily averaged SO$_2$ mixing ratios for the grid cell with the Kilauea volcano ranged from below 0.01 to 0.14 ppbv, while the annual mean value was 0.03 ppbv. In contrast, when volcanic emissions were incorporated in H-CMAQ, simulated SO$_2$ mixing ratios ranged from below 0.03 to 1088.20 ppbv with

an annual mean value of 221.08 ppbv which better matches the dynamic range inferred from the SO$_2$ measurements in the vicinity of the Kilauea volcano reported in Kroll et al. (2015).

      A second location-specific analysis was conducted at site located on Virgin Island and in the state of Florida. As seen in the monthly-average spatial distribution patterns (Fig. 5), the influence of volcanic activity of Soufriere Hills located in Montserrat are more prominent during winter. The nearest observation site from Soufriere Hills is the IMPROVE site VIIS1,

and the southernmost measurement location in Florida (see, Fig. 4) is the CASTNET site EVE419 and comparison of modeled values with observations at these locations are shown in Fig. 6 (b) and (c). At VIIS1 (Fig, 6 (b)), the base H-CMAQ simulation showed invariant concentrations through the year with an annual average value of 0.86 μg/m$^3$. The statistical scores of R showed scattered correspondence between model and observation, NMB showed negative bias because the base H-CMAQ did not capture the episodical peak concentration. By including the degassing volcanic SO$_2$ emissions, the model caught episodical

peak events. The statistical scores showed that the averaged concentration was 1.75 μg/m$^3$, and NMB showed +44.6%, and R was dramatically improved into 0.72. The deterioration in the value of NME was found, because of the continuous model overestimation. As also suggested the case in Hawaii, further refinement on emission treatments is required. At EVE419 (Fig, 6 (c)), the seasonal pattern with summer minima was captured by H-CMAQ but underestimated throughout the year. Increased

concentrations through the incorporation of volcanic emissions are noted during January, late April to early May and December. The increase during late April to early May were not seen from the monthly-averaged spatial distribution (Fig. 5); hence these could be the episodic long-range transport by southeasterly winds. Because of these increased concentrations, all statistical scores of R, NMB, NME showed improvement compared to the base H-CMAQ. CASTNET also provides measurements of weekly-average SO$_2$ mixing ratios, which are used to evaluate the model's performance for SO$_2$ prediction, as shown in Table

S1. In addition to domain-mean evaluation, we evaluated SO$_2$ at the EVE419 site in Florida. At this site, the observed annual mean was 0.61 ppbv whereas base H-CMAQ and H-CMAQ with volcanic SO$_2$ emissions both showed 0.77 ppbv. Additionally, the results suggest that the SO$_2$ from volcanic sources was fully oxidized to SO$_4^{2-}$ during long-range transport and there was no direct transport of SO$_2$ itself at this site. At this CASTNET site of EVE419 at Florida, the conversion rate from SO$_2$ to SO$_4^{2-}$ were further examined. This conversion rate is defined as SO$_4^{2-}$/(SO$_4^{2-}$ + SO$_2$), and higher value indicates the well-oxidation

from SO$_2$ to SO$_4^{2-}$. The temporal variation of conversion rate is also plotted in Fig. 6 (b). The conversion rate was lower in cold season and higher in warm season, and this general feature were captured by model. The mean of conversion rate through the year was 0.58 in the observation whereas original H-CMAQ was 0.42. By incorporating degassing volcanic SO$_2$ emissions, the mean of conversion rate increased in 0.43, but still underestimated the observed value. Since data from routine measurement networks are not designed to specifically characterize impacts of volcanic emissions on atmospheric sulfur budgets, these

observations are unable to unambiguously quantify any modulation in S(IV) to S(VI) conversion rates due to the presence of volcanic emissions as also evidenced by the small change in the estimated conversion rate between the simulations with and without volcanic degassing emissions. Collectively, these results and those summarized in Table 2 and Figure 4 suggest that inclusion of degassing volcanic emissions moderately improve model performance statistics for simulated SO$_4^{2-}$ spatial distributions and temporal variations. Note that the incorporation of volcanic degassing emissions does not necessarily improve

model performance in all instances due to the presence of uncertainties and potentially compensating errors in other parts of the modeling system (i.e. model input fields like other emissions and meteorology, model representation of processes such as chemical reactions and deposition) and suitability of the measurement network (e.g., proximity) in characterizing the variability in tropospheric composition due to volcanic degassing emissions . Nonetheless, incorporating the degassing volcanic SO$_2$ emissions which clearly occur in nature is an important aspect for enhancing the completeness of the modeling system itself.

Quantifying the space and time variations of persistent volcanic degassing emissions on tropospheric composition is important, especially in context of using models to characterize background pollution levels and anthropogenic enhancements of tropospheric pollutants and associated health and visibility impacts.

## 3.4 Impact on upper troposphere

Finally, the impacts caused by including the degassing volcanoes were investigated throughout the troposphere. In subsequent analysis, the boundary layer is defined from the surface to 750 hPa, and the free troposphere is defined from 750 to 250 hPa and pressure levels of 750, 500, and 250 hPa are used to refer as the bottom, middle, and top of the free troposphere similar to our previous study (Itahashi et al., 2020b). The vertical profile of simulated $SO_4^{2-}$ concentration averaged along the edge of western U.S.A. (defined in Fig. 4 (b)) were analyzed. The vertical profile of the base H-CMAQ and the increased concentration due to incorporating the degassing volcanic $SO_2$ emission are plotted in Fig. 7. Modeled $SO_4^{2-}$ concentrations decreased from the surface to upper layers, and concentration levels were greater in summer than those during winter (e.g., Fig. 4 for surface results). Incorporation of volcanic $SO_2$ emissions, increased monthly mean $SO_4^{2-}$ within the boundary layer by up to 0.4 µg/m³ during June to September. This increase is consistent with the spatial distribution at surface depicted in Fig. 5, but also occurs through the depth of the boundary layer. During other months, the simulated concentration increases were less than 0.1 µg/m³ throughout the free troposphere.

Simulated annual average $SO_4^{2-}$ concentration distributions at the bottom, middle and top of the free troposphere from the two model simulations and the estimated increase due to incorporation of degassing volcanic emissions are shown in Fig. 8. The simulated spatial distribution of $SO_4^{2-}$ exhibited similar patterns at the bottom of the free troposphere (Fig. 8 (a), (d)) with higher concentration over Asian region. However, significant differences in the magnitude and spatial distributions of simulated $SO_4^{2-}$ between the two simulations are noted at the middle and top of the free troposphere as seen from higher concentration in the northern Pacific region close to the north pole (Fig. 8 (b), (c), (e), (f)). The maximum concentrations at the top of the free troposphere (Fig. 8 (c), (f)) decreased by a factor of 10 compared to those at the surface (Fig. 3 (a), (b)). The increased concentration by including degassing volcanoes showed the increment corresponded to the location of volcanoes. In contrast to the increased concentrations over Pacific Rim region found at the surface level (Fig. 3 (c)), increased $SO_4^{2-}$ over the Bering Sea was noticeable at the bottom of the free troposphere (Fig. 8 (g)). Over this area, increased $SO_4^{2-}$ ranged from +0.08 to +0.2 at the bottom of free troposphere were found. This is mainly caused by the Mutnovsky, Gorely, Kliuchevskoi, and Bezymianny volcanoes (No. 5 and No. 6 in Table 1) and other volcanoes located on the Kamchatca Peninsula (Nos. 12, 14, 16, 19, 24, 35, 36, and 48), and volcanoes over Alaska Peninsula (Nos. 37, 38, 39, 40, and 46) surrounding Bering Sea. These volcanoes were characterized by the relatively higher altitude (No. 6 was allocated into the highest altitude) considered in this study (Table 1 and Fig. 2). The increased concentration at the middle and top of free troposphere were widely distributed with centering on the north pole. These increased concentrations indicated that while volcanic emissions were assumed to be injected into the lower and middle free troposphere (Fig. 2), their impacts on secondary $SO_4^{2-}$ production were detected throughout the free troposphere. Based on the percentage change relative to the original H-CMAQ simulation, Fig. 8 further shows that the spatial extent of the impact of including degassing volcanic $SO_2$ emissions on $SO_4^{2-}$ increased with height, with most of the northern hemisphere showing increases exceeding 10% at 250 hPa.

**4 Conclusions**

Previous work on the development and evaluation of the H-CMAQ model has historically not considered volcanic $SO_2$ emissions. In this study, satellite-constrained $SO_2$ emissions from 50 degassing volcanoes over the northern hemisphere are incorporated into H-CMAQ, and their impacts on tropospheric $SO_4^{2-}$ is evaluated for model simulations of the calendar year 2010. Model performance was evaluated using network observations of EANET over Asia and CASTNET and IMPROVE over U.S.A. The base H-CMAQ underestimated $SO_4^{2-}$, and this was partially rectified by incorporating volcanic $SO_2$ emissions as indicated through improvements in the statistical scores of R, NMB, and NME. Over Asia, largest increase in $SO_4^{2-}$ were simulated in vicinity of the volcanos, especially over Japan and Indonesia. Over the U.S.A., the largest impact over the central Pacific was caused by including the Kilauea volcano and its impacts on the continental U.S.A. was limited to the western U.S.A. during the summer. The emissions from the Soufriere Hills volcano located in Montserrat affected the southern U.S.A. during the winter season. Analysis of temporal variations at measurement sites located in Hawaii, Virgin Island, and Florida showed improvements in modeling performance as a result of inclusion of volcanic $SO_2$ emissions. The changes in simulated aloft $SO_4^{2-}$ concentration were investigated, and it was revealed that the impact were detected up to the top of the free troposphere although the considered $SO_2$ emissions were injected in the lower and middle of the free troposphere.

The results suggest that emissions of $SO_2$ resulting from volcanic degassing is one of the important sources that should be considered in the modeling system as it regulates tropospheric $SO_4^{2-}$ levels not only in the vicinity of the volcano but also through long-range transport that modulates background levels at downwind continents. In addition to inclusion of persistent degassing emissions, eruptive volcanic emissions can also play an important role in episodically modulating tropospheric sulfur budgets and radiative forcing (Schmidt et al., 2012; 2018). As seen from the analysis of model and measured values at the Kilauea volcanic site in this study, the approach to treat the degassing volcanic $SO_2$ emissions as assigned in one model vertical layer without any temporal variation sometimes lead to the unexpected positive bias on cleaner days. A more realistic treatment of temporal variation of degassing gasses and the related meteorological parameters (e.g., local wind speed) would be useful to further refine the modeling performance of $SO_4^{2-}$ produced via volcanic $SO_2$ emission.

**Code availability**

The CMAQ version 5.2 are available from https://doi.org/10.5281/zenodo.1167892 (United States Environmental Protection Agency, 2017).

**Data availability**

The observational datasets for ambient concentration used in this study are available from their respective websites: http://www.eanet.asia/index.html (EANET), https://www.epa.gov/castnet (CASTNET), and http://vista.cira.colostate.edu/Improve (IMPROVE) for surface observation network. Last Access: 31 August 2020. The

observational datasets of concentration in precipitation and wet deposition used in this study are available from http://nadp.slh.wisc.edu (NADP/NTN). Last Access: 30 July 2021 The satellite observation of $SO_2$ column by OMI used in this study are taken from https://doi.org/10.5067/Aura/OMI/DATA3008. Last Access: 30 July 2021. The emissions for original H-CMAQ simulation and degassing volcanic $SO_2$ emissions established in this study can be available upon the request.

**Competing interests**

The authors declare that they have no conflict of interest.

**Disclaimer**

10  The views expressed in this paper are those of the authors and do not necessarily reflects the views or policies of the U.S. Environmental Protection Agency.

**Author contributions**

Syuichi Itahashi performed the incorporation of degassing volcanic $SO_2$ emissions, analysis of observations and model 15  simulation results, and prepared the manuscript with contributions from all co-authors. Rohit Mathur, Christian Hogrefe, and Sergey L. Napelenok contributed to establish the hemispheric modeling application for this study and prepared the original emission dataset and initial condition from previous long-term simulation results. Yang Zhang contributed to the literature review and refined this research through simulation designs, and result analyses and interpretation.

20  **Acknowledgement**

We thank Golam Sarwar and Barron Henderson for constructive suggestions on the initial version of this manuscript. The authors are grateful for the available observation dataset (EANET, IMPROVE, and CASTNET). Yang Zhang acknowledges support from the 2019-2020 North Carolina State University's Kelly Memorial Fund for US-Japan Scientific Cooperation.

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

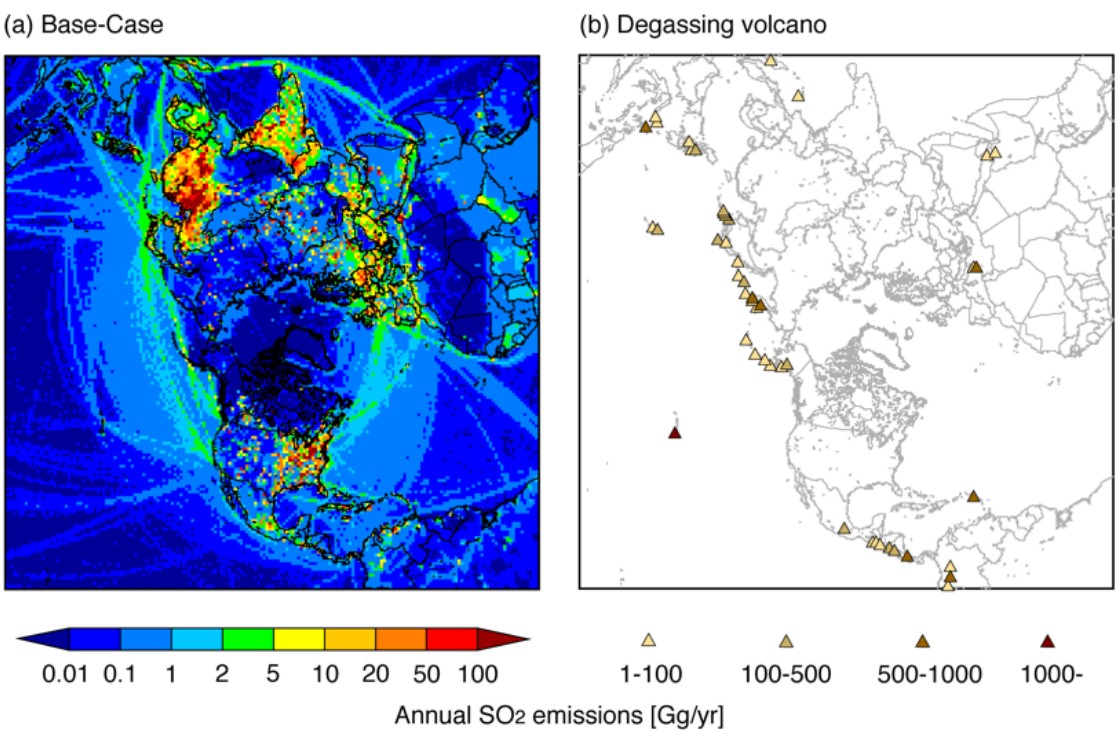

**Figure 1. Geographical mapping of the (a) original SO₂ emissions used in H-CMAQ and (b) degassing volcanic annual SO₂ emissions incorporated in this study during 2010.**

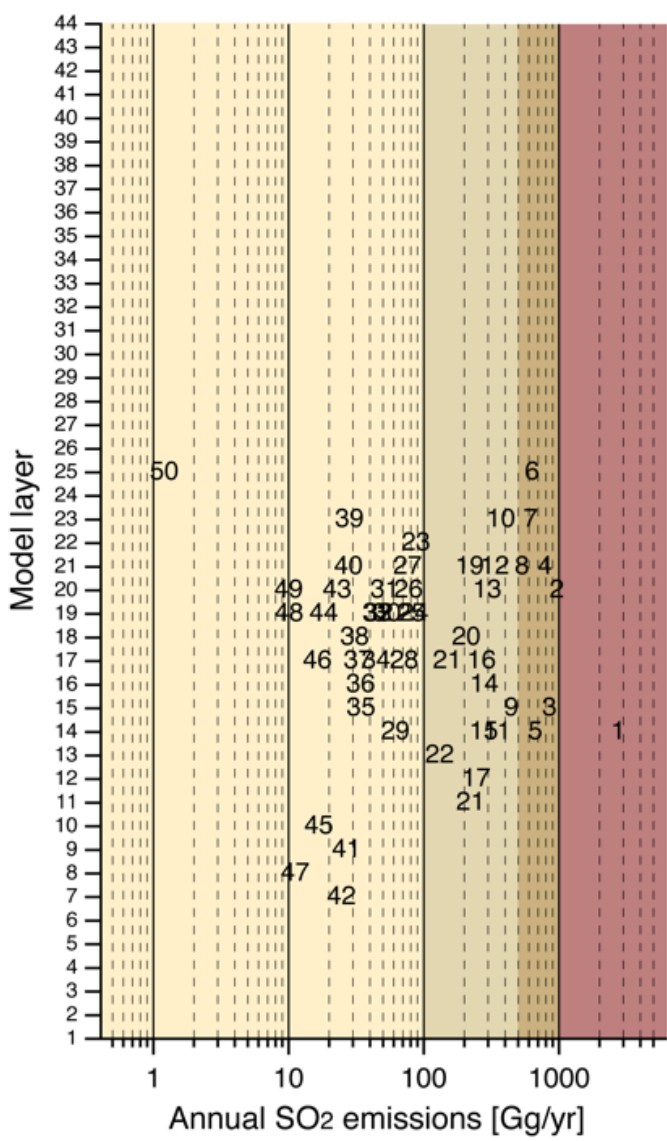

**Figure 2. The model layers to which the SO₂ emissions from the 50 volcanoes considered in this study were assigned. See Table 1 for a listing of the volcanoes represented by each number in this Figure. In Table 1, volcanoes are listed in decreasing order of their annual SO₂ emission amount.**

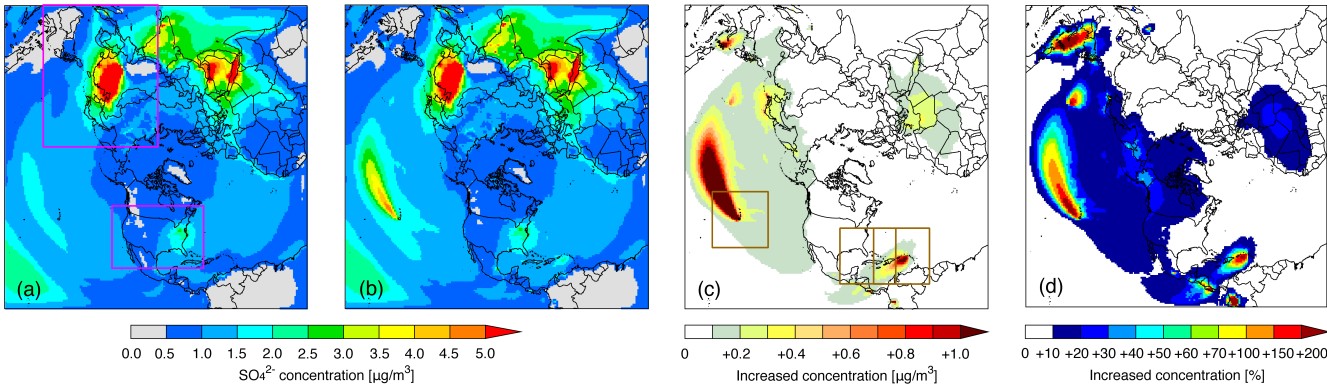

**Figure 3. Simulated annual averaged SO₄²⁻ concentration in 2010 by (a) original H-CMAQ and (b) H-CMAQ with incorporation of volcanic SO₂ emissions, (c) increased concentration by the incorporation of volcanic SO₂ emissions, and (d) same as (c) but shown as relative percentages at the surface. The rectangular regions colored in pink in panel (a) indicate the Asia (left-top) and U.S.A. (right-bottom) subdomains used for detailed analysis. The rectangular regions colored in brown in panel (c) indicate areas for the analysis on specific site analysis where largely impacted from degassing SO₂ volcanoes.**

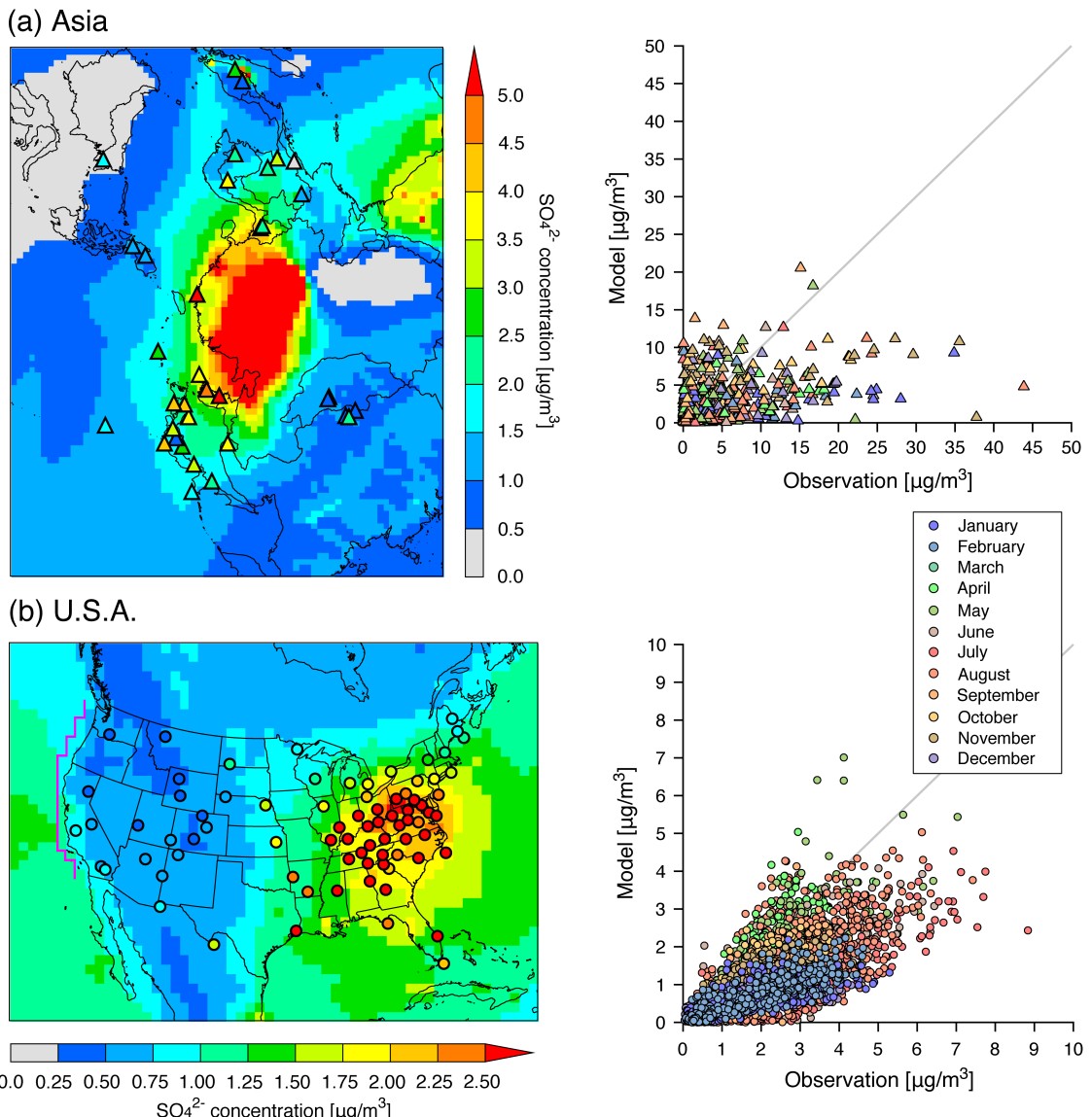

**Figure 4. Simulated annual averaged SO$_4^{2-}$ concentration in 2010 by original H-CMAQ over (a) Asia and (b) U.S.A. with overlaid EANET and CASTNET surface observations. Scatter-plots between surface observation (EANET over Asia, and CASTNET over U.S.A.) and original H-CMAQ with identification colors for each month is also shown. Note that the color-scale is different for Asia and U.S.A. The pink line over the western U.S.A. in panel (b) is the defined western edge to analyze the vertical profile of SO$_4^{2-}$ concentrations (see, Fig. 7).**

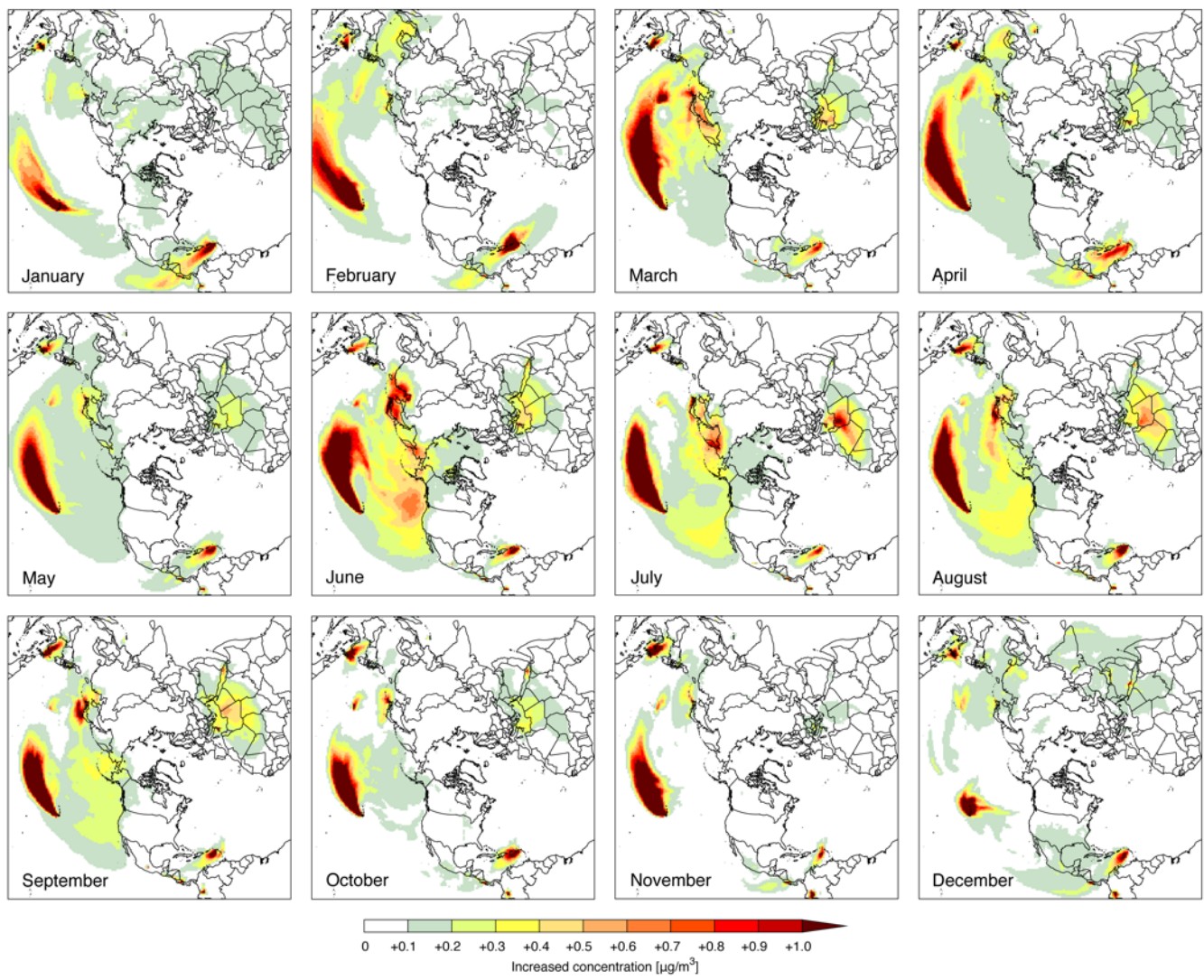

**Figure 5. Simulated increased surface SO$_4^{2-}$ concentration by the incorporation of volcanic SO$_2$ emissions over each month in 2010 at surface.**

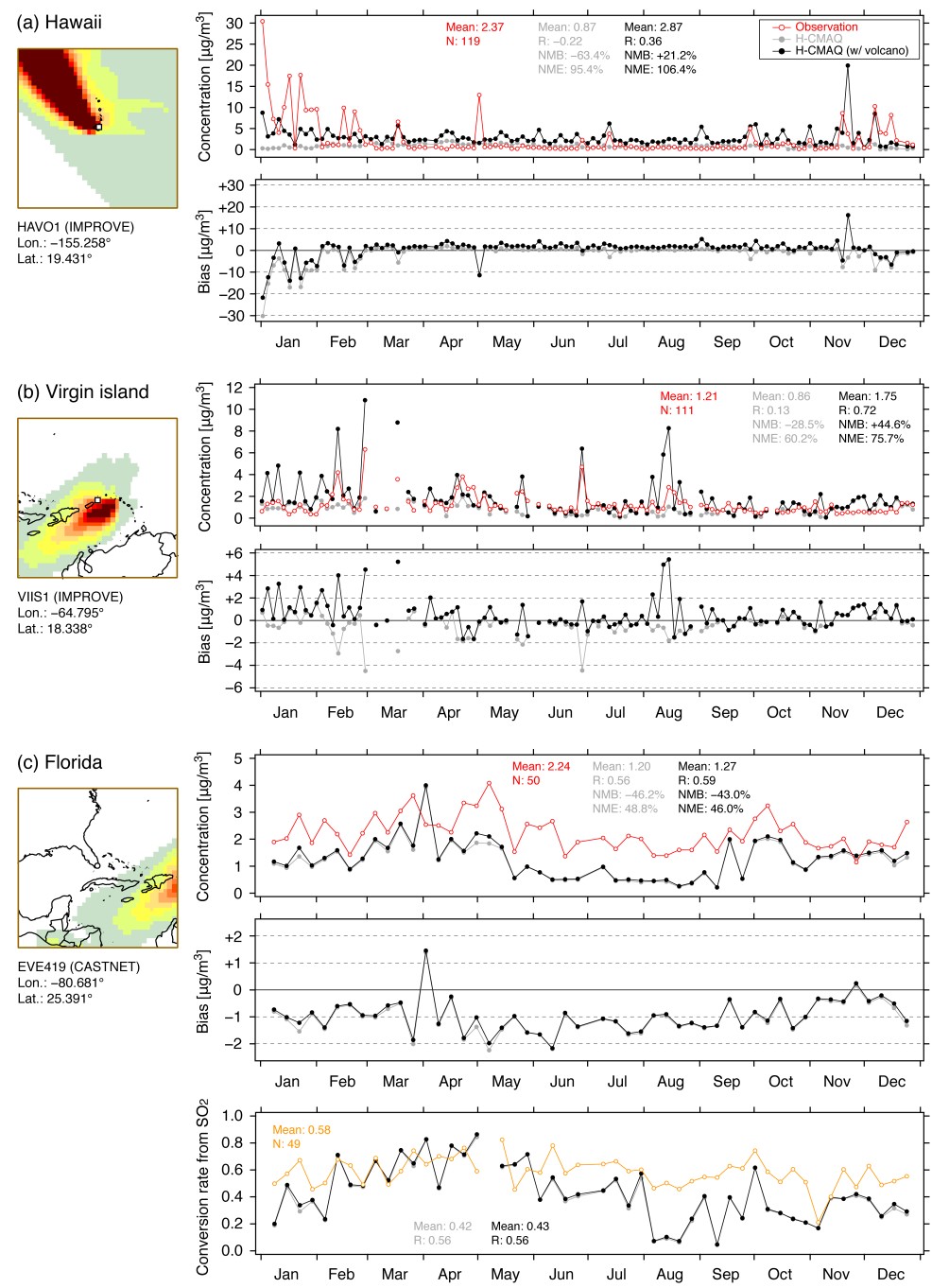

**Figure 6. Temporal variation of observed and simulated SO₄²⁻ concentrations and simulation biases at (a) Hawaii IMPROVE site (HAVO1), (b) Virgin Island IMPROVE site (VIIS1), and (c) Florida CASTNET site (EVE419) in 2010. Red open circle denotes observations, and gray and black closed circles denote original H-CMAQ and H-CMAQ incorporating volcanoes, respectively. Statistical scores are indicated in the inset. The comparison of conversion rate from SO₂ to SO₄²⁻ are also shown in (c).**

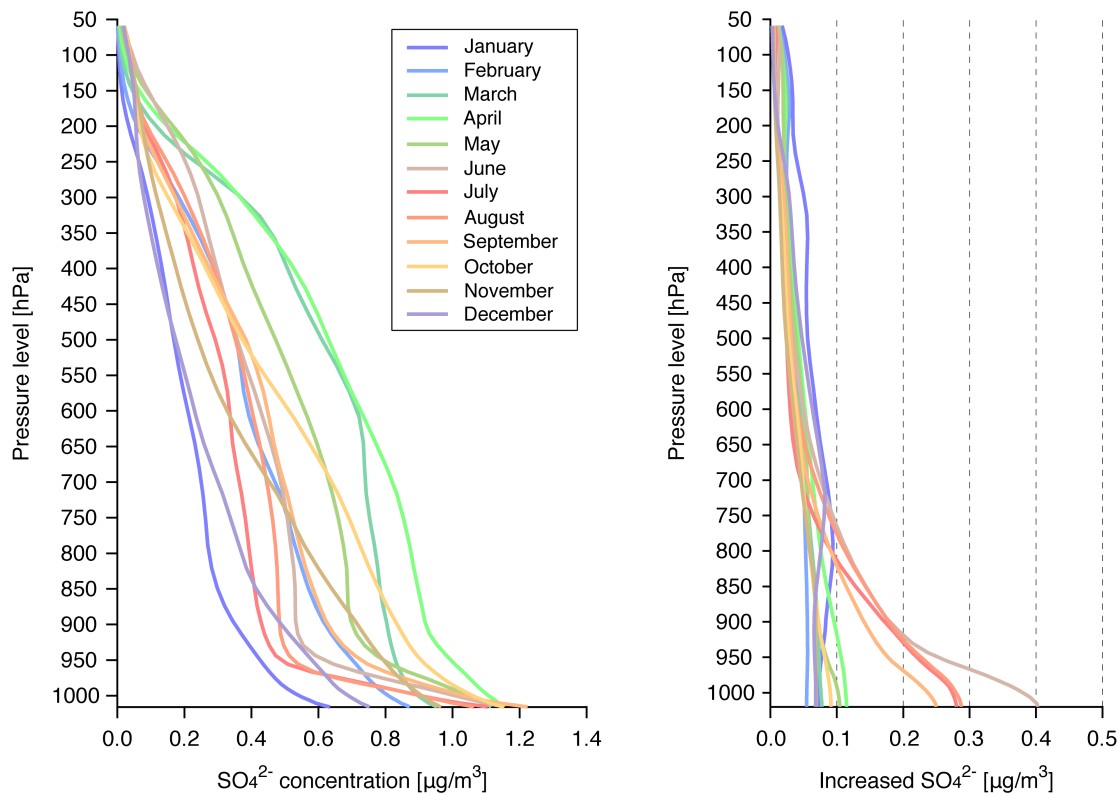

**Figure 7. Vertical profile of simulated SO₄²⁻ concentration of each month in 2010 at the edge of western U.S.A. (see, Fig. 4) (left) original H-CMAQ and (right) increased concentration by the incorporation of volcanic SO₂ emissions.**

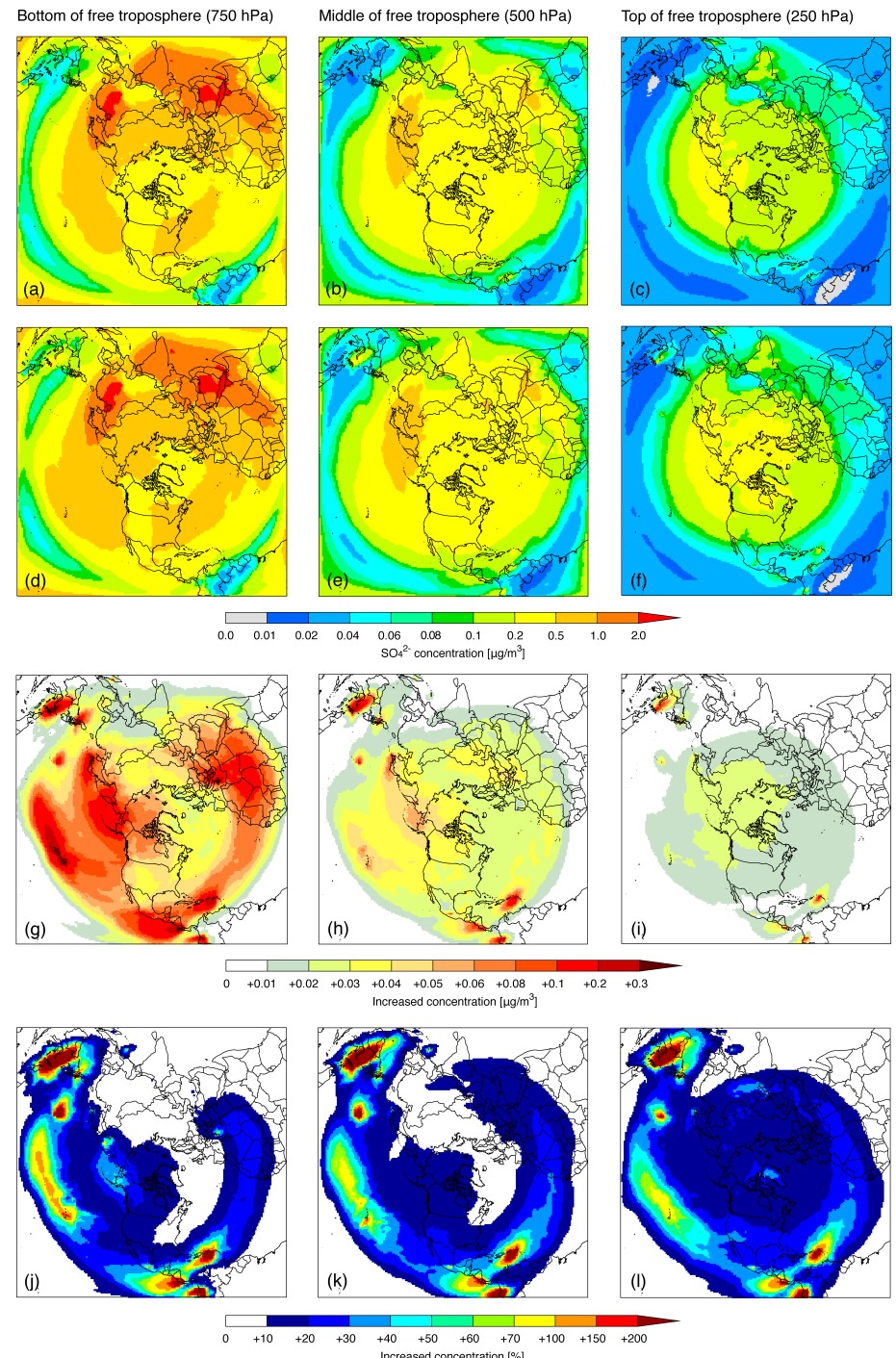

**Figure 8. Simulated annual averaged SO$_4^{2-}$ concentration in 2010 by (a-c) original H-CMAQ and (d-f) H-CMAQ with incorporation of volcanic SO$_2$ emissions, (g-i) increased concentration by the incorporation of volcanic SO$_2$ emissions, and (j-l) same as (g-i) but shown as relative percentage at the bottom of the free troposphere (750 hPa; a, d, g, j), middle of the free troposphere (500 hPa; b, e, h, k), and top of the free troposphere (250 hPa; c, f, I, l).**

**Table 1. Information of volcanoes incorporated in this study.**

| No. | Name | Country | Longitude [°] | Latitude [°] | Altitude [m a.s.l] | Emission [Gg/yr] |
|---|---|---|---|---|---|---|
| 1 | Kilauea | USA | -155.29 | 19.42 | 1222.0 | 2739.0 |
| 2 | Mutnovsky + Gorely | Russia | 158.20 | 52.45 | 2322.0 | 960.4 |
| 3 | Dukono | Indonesia | 127.88 | 1.68 | 1170.0 | 848.9 |
| 4 | Turrialba + Poas | Costa Rica | -83.77 | 10.03 | 3340.0 | 788.5 |
| 5 | Soufriere Hills | Montserrat (UK) | -62.18 | 16.72 | 870.0 | 667.1 |
| 6 | Kliuchevskoi + Bezymianny | Russia | 160.64 | 56.06 | 4835.0 | 631.2 |
| 7 | Nevado del Huila | Colombia | -76.03 | 2.93 | 5364.0 | 617.6 |
| 8 | Etna | Italy | 15.00 | 37.73 | 2711.0 | 533.3 |
| 9 | Sakura-jima | Japan | 130.65 | 31.59 | 1117.0 | 443.5 |
| 10 | Popocatepetl | Mexico | -98.62 | 19.02 | 5100.0 | 372.1 |
| 11 | Suwanose-jima | Japan | 129.71 | 29.64 | 796.0 | 349.3 |
| 12 | Shiveluch | Russia | 161.34 | 56.64 | 3283.0 | 334.7 |
| 13 | Mayon | Philippines | 123.69 | 13.26 | 2462.0 | 293.7 |
| 14 | Karymsky | Russia | 159.45 | 54.05 | 1536.0 | 280.2 |
| 15 | Miyake-jima | Japan | 139.53 | 34.09 | 775.0 | 276.7 |
| 16 | Sarychev | Russia | 153.21 | 48.08 | 1200.0 | 267.3 |
| 17 | Pagan | Northern Mariana Islands | 145.79 | 18.14 | 570.0 | 244.5 |
| 18 | Masaya | Nicaragua | -86.16 | 11.98 | 635.0 | 221.3 |
| 19 | Avachinsky | Russia | 158.83 | 53.25 | 2741.0 | 218.1 |
| 20 | San Cristobal +Telica | Nicaragua | -87.00 | 12.70 | 1745.0 | 206.0 |
| 21 | Aso | Japan | 131.10 | 32.88 | 1592.0 | 149.2 |
| 22 | Satsuma-Iojima | Japan | 130.31 | 30.79 | 704.0 | 131.0 |
| 23 | Shishaldin | USA | -163.97 | 54.76 | 2857.0 | 87.4 |
| 24 | Chikurachki + Ebeko | Russia | 155.46 | 50.33 | 1816.0 | 85.0 |
| 25 | Asama | Japan | 138.52 | 36.41 | 2568.0 | 80.3 |
| 26 | Fuego + Pacaya | Guatemala | -90.88 | 14.47 | 3763.0 | 77.0 |
| 27 | Redoubt | USA | -152.75 | 60.49 | 3108.0 | 75.7 |
| 28 | Bulusan | Philippines | 124.05 | 12.77 | 1500.0 | 71.5 |
| 29 | Stromboli | Italy | 15.21 | 38.79 | 870.0 | 61.6 |
| 30 | Karangetang | Indonesia | 125.40 | 2.78 | 1780.0 | 53.3 |
| 31 | Santa Maria | Guatemala | -91.55 | 14.76 | 3772.0 | 51.5 |
| 32 | Sinabung | Indonesia | 98.39 | 3.17 | 2460.0 | 46.4 |
| 33 | Tokachi | Japan | 142.69 | 43.42 | 2077.0 | 45.0 |
| 34 | Lokon-Empung | Indonesia | 124.79 | 1.36 | 1580.0 | 44.4 |
| 35 | Ketoi | Russia | 152.48 | 47.34 | 870.0 | 34.3 |
| 36 | Kudriavy | Russia | 148.84 | 45.39 | 1125.0 | 34.1 |
| 37 | Augustine | USA | -153.45 | 59.35 | 1252.0 | 32.8 |
| 38 | Gareloi | USA | -178.79 | 51.79 | 1573.0 | 30.9 |
| 39 | Spurr | USA | -152.25 | 61.30 | 3374.0 | 28.2 |
| 40 | Veniaminof | USA | -159.38 | 56.17 | 2507.0 | 27.7 |
| 41 | Anatahan | Northern Mariana Islands | 145.67 | 16.35 | 320.0 | 26.8 |
| 42 | Barren Island | India | 93.86 | 12.28 | 230.0 | 24.6 |
| 43 | Galeras | Colombia | -77.39 | 1.20 | 4276.0 | 22.8 |
| 44 | Santa Ana | El Salvador | -89.63 | 13.85 | 2381.0 | 18.2 |
| 45 | Alu-Dalafilla + Erta Ale | Ethiopia | 40.67 | 13.60 | 613.0 | 16.7 |
| 46 | Cleveland | USA | -169.77 | 52.83 | 1170.0 | 16.3 |

| 47 | Jebel-at-Tair | Yemen | 41.83 | 15.55 | 244.0 | 11.2 |
| 48 | Kizimen | Russia | 160.36 | 55.12 | 2376.0 | 10.1 |
| 49 | Kanlaon | Philippines | 123.13 | 10.41 | 2435.0 | 10.0 |
| 50 | Nevado del Ruiz | Colombia | -75.32 | 4.90 | 5321.0 | 1.2 |

**Table 2. Statistical analysis of modeled SO$_4^{2-}$ concentration with observations.**

| | N | Mean | | R | NMB | NME |
|---|---|---|---|---|---|---|
| | | Observation | Model | | | |
| EANET | | | | | | |
| −original H-CMAQ | 1167 | 3.68 | 2.30 | 0.43[***] | −37.6% | 67.0% |
| −incorporation of volcanic emissions | | | 2.44 | 0.43[***] | −33.6% | 66.0% |
| CASTNET | | | | | | |
| −original H-CMAQ | 4216 | 1.91 | 1.36 | 0.73[***] | −28.7% | 39.4% |
| −incorporation of volcanic emissions | | | 1.42 | 0.73[***] | −25.6% | 38.6% |
| IMPROVE | | | | | | |
| −original H-CMAQ | 18844 | 1.05 | 0.94 | 0.64[***] | −10.6% | 53.9% |
| −incorporation of volcanic emissions | | | 1.02 | 0.64[***] | −3.2% | 46.0% |

Note: The unit of mean for observations and simulations is μg/m$^3$. Suggested threshold values of R > 0.70, NMB < ±10%, and NME < 35% as performance goal, and threshold values of R > 0.40, ±10% < NMB < ±30%, and 35% < NME < 50% as performance criteria for daily SO$_4^{2-}$ by the regional-scale modeling reviewed by Emery et al. (2017). Significance levels by Students' t-test for correlation coefficients between observations and simulations are remarked as [*]$p < 0.05$, [**]$p < 0.01$, and [***]$p < 0.001$, and lack of a mark indicates no significance.