# Peer review of "Incorporation of volcanic SO2 emissions in the Hemispheric CMAQ (H-CMAQ) version 5.2 modeling system and assessing their impacts on sulfate aerosol over Northern Hemisphere"

_Geoscientific Model Development, 2021_

## Author Comment (AC1)

Response to CEC:

Dear authors,

After checking your manuscript, it has come to our attention that it does not comply with our Code and Data Policy.

https://www.geoscientific-model-development.net/policies/code_and_data_policy.html

You have archived your code in GitHub. However, GitHub is not a suitable repository. GitHub itself instructs authors to use other alternatives for long-term archival and publishing, such as Zenodo. Therefore, please, publish your code in one of the appropriate repositories, and include the relevant primary input/output data. In this way, you must include in a potential reviewed version of your manuscript the modified 'Code and Data Availability' section, the DOI of the code (and another DOI for the dataset if necessary).

Also, in the GitHub repository it says that the code is "open-source"; however, there is no license listed. If you do not include a license, despite what you state in the README file, the code is not "open-source", it continues to be your property. Therefore, when uploading the model's code to Zenodo, you could want to choose a free software/open-source (FLOSS) license. We recommend the GPLv3. You only need to include the file 'https://www.gnu.org/licenses/gpl-3.0.txt' as LICENSE.txt with your code. Also, you can choose other options that Zenodo provides: GPLv2, Apache License, MIT License, etc.

Juan A. Anel

Geosc. Mod. Dev. Exec. Editor

**Reply:**

**We appreciate your notice on our statement for Code and Data availability. We reviewed the policy of this journal, and fully revised the code availability statement as follows.**

**Code availability**

The CMAQ version 5.2 are available from https://doi.org/10.5281/zenodo.1167892 (United States Environmental Protection Agency, 2017).

We also added the appropriate reference as follows.
Reference:

United States Environmental Protection Agency: CMAQ (Version 5.2) [Software], Zenodo, https://doi.org/10.5281/zenodo.1167892, 2017.

We believe that these statements can comply the data policy of the journal of *Geoscientific Model Development.*

---

## Author Comment (AC2)

Response to the Referee #1:

This paper described a straightforward sensitivity study of volcanic SO2 emission using the hemispheric CMAQ. It conducted two runs, with and without the volcanic SO2 emissions, and the results were mainly compared to surface sulfate measurements for year 2010. This surface sulfate-only verification is not sufficient for volcanic SO2 emissions since that sulfate concentration can be affected by other processes, such as wet scavenging. You may need to compare the modeled SO2 concentrations to surface/aircraft measurements and satellite retrievals. This manuscript did not mention the temporal variations of volcanic SO2 emission used here, and it likely used static emission rates. If so, the corresponding discussions are needed to justify the treatment since the volcanos unlikely erupted at constant rates for whole year of 2010.

**Reply:**

**We appreciate the reviewer's constructive comments. To address the reviewer's concerns on model evaluation for SO₂, we have significantly expanded the model evaluation discussion to now also include (1) comparisons of column SO₂ predictions for both model cases – with and without volcanic degassing emissions, with OMI inferred SO₂ column distributions across our Northern Hemisphere domain (in Figure S1 in the Supplement), and (2) comparison of ambient SO₂ predictions with observations at CASTNET monitors in the U.S. (Table S1), and (3) comparison of model predicted SO₄²⁻ concentrations in rainwater, precipitation amounts, and SO₄²⁻ wet deposition amounts with observations from the NADP in the US and EANET in Asia (Table S2). Incorporation of volcanic degassing emissions in the model generally helps improve the performance metrics (slightly in some cases) in these model-observation comparisons and are detailed further in the revised manuscript discussion.**

**In terms of the temporal variation of SO$_2$ emission, due to lack of other information, we used a time-invariant emission rate. For clarity, we have revised Section 2.1 as follows with underlined for additional discussion.**

**"In this study, annual estimates of degassing volcanic SO$_2$ emissions by Carn et al. (2017) are incorporated into H-CMAQ. The estimated emission of SO$_2$ from the 50 volcanos within our northern hemisphere modeling domain (Figure 1b) is 12.7 Tg/yr. Considering the characteristics of degassing process from volcanoes and the lack of any other information to accurately specify its temporal variations, we use a constant SO$_2$ emission rate in H-CMAQ based on the annual SO$_2$ emission estimates by Carn et al. (2017)."**

**Please also see our replies to your specific comments.**

Specified comments:

Page 4, line 24: "In this study, the entire year of 2010 was simulated". Why choose 2010 as the studied year, or is there any specific reason related to the 2010 volcano eruptions?

**Reply:**

**As already stated in the introduction section, we aimed to understand the impact of SO$_2$ emissions from degassing volcanoes in this study and did not focus on any specific volcanic eruption event.**

**P2, L32-33: "The volcanic SO$_2$ emissions from degassing are relatively stable at 23.0±2.3 Tg-SO$_2$/yr during 2005-2015, and the highest amount was approximately 26 Tg-SO$_2$/yr in 2010."**

**P3, L5-7: "According to the comparison between the degassing and eruptive emissions, Carn et al. (2017) estimated that during 2005-2015, volcanic activity contributed about 23 Tg/yr of SO$_2$ due from degassing while eruptive SO$_2$ emissions ranged from 0.2 to 10 Tg/yr of SO$_2$. Therefore, understanding the behavior of persistent degassing SO$_2$ emissions and its contributions to airborne SO$_4^{2-}$ levels is important."**

**Based on these reasons, we choose 2010 as the studied year. To address the reviewer's comment, we stated the reason for selection of 2010 explicitly in page 4, lines 23-26:**

**"In this study, the entire year of 2010 was simulated to analyze the volcanic emission impacts over the Northern Hemisphere, as the $SO_2$ emissions from degassing during the 2005-2015 period were highest in 2010. Also note that emission estimates presented in Carn et al. (2017) suggest that degassing dominate over eruptive $SO_2$ emissions during the same time period."**

Page 5, line 7. So the volcano emissions have no plume rise, right? If so, why?

**Reply:**

**Yes, plume rise for volcanic degassing emissions were not treated in this study, and this was because the characteristics of the degassing process. As noted earlier, we aimed to analyze the impact of $SO_2$ emissions from persistent degassing volcanoes and did not focus on any specific volcanic eruption. As such, a plume rise calculation would require characterization of buoyancy of the degassing plume and thus need information (either observations or estimates) on heat content of the degassing plume. Since these are not readily available, volcanic degassing $SO_2$ emissions were allocated into the model vertical layer corresponding to the volcano's altitude. In this configuration, the 108 km grid spacing used in CMAQ does not allow the model to adequately resolve localized terrain peaks such as volcanoes and assigning their emissions to the first model layer would not account for the fact that in reality these emissions typically occur above the mixed layer. Therefore, the vertical layer to which volcanic degassing emissions were assigned was determined by first calculating the difference between the altitude of a given volcano (this was listed in Table 1) and the CMAQ terrain height for the cell in which it is located, and then determining the vertical CMAQ layer corresponding to this difference.**

Page 6, section 3.1. As commented above, the verification with only surface sulfate is insufficient. Even with the coarse resolution, the SO2 comparisons are still preferred. Or, you can use a high-resolution regional CMAQ to study certain region for a certain period.

**Reply:**

**We agree that comparisons with $SO_2$ measurements would enable more direct assessments of the impacts of including volcanic $SO_2$ emissions.**

**CASTNET has been routinely monitoring $SO_2$ concentration. To address the reviewer's comment, we evaluated $SO_2$ predictions against all available observations from CASTNET. The evaluation results are now newly presented in Table S1 in the Supplement along with brief discussions of these results in the main text.**

**Table S1. Statistical analysis of modeled $SO_2$ concentration against CASTNET observations.**

| | N | Mean | | R | NMB | NME |
|---|---|---|---|---|---|---|
| | | Obs. | Model | | | |
| **CASTNET** | | | | | | |
| **−original H-CMAQ** | **4216** | **1.69** | **2.81** | **0.57***** | **+66.1%** | **94.7%** |
| **−incorporation of volcanic emissions** | | | **2.83** | **0.58***** | **+67.4%** | **94.9%** |

**Note: The unit of mean for observations and simulations is ppbv. Significance levels by Students' t-test for correlation coefficients between observations and simulations are remarked as $^*$p < 0.05, $^{**}$p < 0.01, and $^{***}$p < 0.001, and lack of a mark indicates no significance.**

**Because $SO_2$ concentration did not change in the case of the incorporation of volcanic emissions, this result showed that $SO_2$ from volcanic sources was fully oxidized to $SO_4^{2-}$ and there was no direct transport of $SO_2$ itself. To indicate this point, we have explicitly stated within the modeling evaluation at specific site of Florida in Section 3.3, and it was revised as follows.**

**"CASTNET also provides measurements of weekly-average SO₂ mixing ratios, which are used to evaluate the model's performance for SO₂ prediction, as shown in Table S1. In addition to domain-mean evaluation, we evaluated SO₂ at the EVE419 site in Florida. At this site, the observed annual mean was 0.61 ppbv whereas base H-CMAQ and H-CMAQ with volcanic SO₂ emissions both showed 0.77 ppbv."**

Page 8-9, section 3.3. The volcanic SO2 impact is only shown at two surface sites and for sulfate only, which is insufficient.

**Reply:**

**We appreciate this suggestion. It should be noted that we have evaluated model performance for airborne $SO_4^{2-}$ concentration relative to all locations available from the EANET, CASTNET, and IMPROVE networks as listed in Table 2. Within this comparison, total of 1167, 4216, and 18844 observation-prediction data pairs have been evaluated. In addition, according to your comment, we have also evaluated model performance for ambient SO₂ concentration for CASTNET observation as listed in Table S1.**

**Taking into account this comment, we have re-examined the analysis in Figure 6, and have revised it to include analysis at an additional observation site in the Virgin Islands. We have revised Figure 6 to show detailed analysis over three specific sites that were most affected by the incorporation of persistent volcanic degassing emissions. The updated discussion in Section 3.3 of the revised manuscript is as follows.**

**"A second location-specific analysis were conducted at sites located in the Virgin Island and in the state of Florida. As seen in the monthly-average spatial distribution patterns (Fig. 5), the influence of volcanic activity of Soufriere Hills located in Montserrat are more prominent during winter. The nearest observational sites from Soufriere Hills are the IMPROVE site VIIS1, and the southernmost measurement location in Florida (see, Fig. 4) is the CASTNET site**

EVE419 and comparison of modeled values with observations at these locations are shown in Fig. 6 (b) and (c). At VIIS1 (Fig. 6 (b)), the base H-CMAQ simulation showed invariant concentrations throughout the year with an annual average value of 0.86 μg/m$^3$. The statistical scores of R showed scattered correspondence between model and observation. NMB showed negative bias because the base H-CMAQ did not capture the episodical peak concentration. By including the degassing volcanic $SO_2$ emissions, the model exhibited improved skill in representing the episodic peaks. The statistical scores showed that the average concentration was 1.75 μg/m$^3$, and NMB showed +44.6%, and R was dramatically improved to 0.72. The deterioration in the value of NME was found, because of the continuous model overestimation. As also suggested the case in Hawaii, further refinement on emission treatments is required. At EVE419 (Fig. 6 (c)), the seasonal pattern with summer minima was captured by H-CMAQ but underestimated throughout the year. Increased concentrations through the incorporation of volcanic emissions are noted during January, late April to early May and December. The increases during late April to early May were not seen from the monthly-averaged spatial distribution (Fig. 5); hence these could be the episodic long-range transport by southeasterly winds. Because of these increased concentrations, all statistical scores of R, NMB, NME showed improvement compared to the base H-CMAQ."

---

## Author Comment (AC3)

Response to the Referee #2:

This study investigates the degassing volcanic SO2 emissions to global sulfur budgets using a Hemispheric CMAQ. The sensitivity simulations were conducted with and without the volcanic SO2 emissions. The model simulations have been verified by the surface sulfate measurements during 2010 around the world. This study indicated that the degassing volcanic SO2 emissions are an important source impacting airborne sulfur budgets and should be considered in air quality model simulations assessing background sulfate levels and their source attribution. Overall, this is a nice piece of paper with clear objectives and methods. Before considering publication in ACP, major revisions should be made. Some comments and suggestions are listed as follows:

**Reply:**

**We thank the reviewer for the overall positive assessment of the manuscript and appreciate the constructive comments, addressing which have helped improve the manuscript. Detailed below is our responses to specific comments by the reviewer.**

Specific comment:

- In model verification, only the measurement of surface sulfate concentration is not enough. Since this study aimed to the sulfur budgets, the observation of deposition as well as VCD from satellite are needed.

**Reply:**

**We appreciate the reviewer's comments on the need for further analysis of sulfur deposition amounts and VCD. In the revised manuscript, we include a supplemental materials section that illustrates the evaluation of surface $SO_2$ concentration, $SO_2$ VCD with satellite, and $SO_4^{2-}$ wet deposition.**

**For $SO_2$ VCD comparison, the publicly available OMI satellite observation do not include information of the averaging kernel to consider the sensitivity on vertical profile. To overcome this point, our previous study (Itahashi et al., 2017a; see figure below) examined the sensitivity of the modeled VCD to the depth from the center**

of mass altitude (CMA), which set as 0.9 km in the satellite retrieval. The results indicated that the analysis of CMA of 0.9 km with 1 km depth provided the best correspondence between model and satellite observation.

[Figure]

Supporting Figure (taken from Figure S6 of Itahashi et al. (2017)): Spatial distribution of $SO_2$ column density (a) observed by space-based OMI, and modeled with a CMA of 0.9 km and depths of (b) 700, (c) 1000, and (d) 1300 m. Units are Dobson Units.

Following this methodology, we have evaluated H-CMAQ with OMI satellite measurement. Because of this comparison methodology, only R was evaluated for the base H-CMAQ simulation and the H-CMAQ simulation incorporating the degassing volcanic $SO_2$ emissions. The results of this evaluation of $SO_2$ VCD are presented in the Supplement and shown as follows.

[Figure]

**Figure S1. Annual averaged SO₂ column in 2010 of (a) observed by OMI satellite and simulated by (b) original H-CMAQ and (c) H-CMAQ with incorporation of volcanic SO₂ emissions. The spatial correlation coefficient with satellite observation are noted at left-bottom corner of (b) and (c).**

**Consistent with the distribution of SO₂ emissions shown in Fig. 1, high SO₂ VCD were also found over portions of East Asia, India, and the Arabian Peninsula. High SO₂ VCD over Hawaii are also indicative of SO₂ emissions from the Kilauea volcano. The original H-CMAQ did perform well over high SO₂ VCD areas such as East and South Asia with high anthropogenic SO₂ emissions; however, it did not capture the peak at Kilauea because of missing volcanic emissions. The correlation coefficient (R) between the original H-CMAQ and OMI VCD was 0.48. In contrast, the H-CMAQ simulation incorporating degassing volcanic SO₂ emissions captured the high SO₂ VCD at Kilauea, and also yielded a higher correlation of 0.56. These comparisons indicate that the incorporation of degassing volcanic SO₂ emissions is important for improving the model's ability in representing magnitude and spatial variability of airborne sulfur across the Northern Hemisphere.**

**For deposition, observed wet deposition data from EANET over Asia and NADP over the U.S.A. was used to further evaluate the model's performance in**

representing precipitation chemistry and atmospheric sulfur budgets. These comparisons are tabulated in the Supplementl and also shown as below.

Table S2. Statistical analysis of modeled $SO_4^{2-}$ concentration in precipitation and wet deposition with observations.

| | N | Mean | | R | NMB | NME |
|---|---|---|---|---|---|---|
| | | Obs. | Model | | | |
| **$SO_4^{2-}$ concentration in precipitation** | | | | | | |
| **EANET** | | | | | | |
| −original H-CMAQ | 2657 | 4743.7 | 1152.5 | 0.35*** | −75.7% | 81.9% |
| −incorporation of volcanic emissions | | | 1236.0 | 0.35*** | −73.9% | 81.6% |
| **NADP** | | | | | | |
| −original H-CMAQ | 7377 | 842.7 | 631.5 | 0.33*** | −25.1% | 61.9% |
| −incorporation of volcanic emissions | | | 653.1 | 0.32*** | −22.5% | 61.6% |
| **Precipitation** | | | | | | |
| **EANET** | 4497 | 16.6 | 13.7 | 0.37*** | −17.4% | 56.3% |
| **NADP** | 10670 | 21.4 | 16.5 | 0.55*** | −23.1% | 66.9% |
| **$SO_4^{2-}$ wet deposition** | | | | | | |
| **EANET** | | | | | | |
| −original H-CMAQ | 2676 | 406.7 | 122.0 | 0.45*** | −70.0% | 78.7% |
| −incorporation of volcanic emissions | | | 131.5 | 0.44*** | −67.6% | 77.9% |
| **NADP** | | | | | | |
| −original H-CMAQ | 8154 | 160.4 | 99.3 | 0.46*** | −38.1% | 68.5% |
| −incorporation of volcanic emissions | | | 103.6 | 0.47*** | −35.4% | 68.4% |

Note: The unit of mean for observations and simulations is g/L for concentration in precipitation and g/ha for wet deposition. Significance levels by Students' t-test for correlation coefficients between observations and simulations are remarked as *p < 0.05, **p < 0.01, and ***p < 0.001, and lack of a mark indicates no significance.

The original H-CMAQ underestimated both $SO_4^{2-}$ concentrations in precipitation and $SO_4^{2-}$ wet deposition. The inclusion of degassing volcanic $SO_2$ emission helped slightly rectify the negative model bias noted in the base calculation.

The results are presented in the Supplement as one figure and two tables and the relevant discussion was added in the main text. The additional text are as follows. In Section 2.2, the explanation of additional observation are added:

[revised manuscript text omitted]

- Since the atmospheric sulfate is connected with nitrate as well as ammonium, analysis on these two species are also needed, especially considering the long-range transport over the remote areas.

  **Reply:**

  **We appreciate this comment to evaluate other aerosol components such as $NO_3^-$ and $NH_4^+$. While we acknowledge the connections of the $SO_4^{2-}$-$NO_3^-$-$NH_4^+$ system and that the gas-particle partitioning of airborne $NH_x$ and total nitrate is dependent on the amounts of airborne $SO_4^{2-}$, we do not believe that biases/errors in predictions of $NO_3^-$ and $NH_4^+$ provide any additional information on the error/bias in $SO_4^{2-}$ predictions than direct comparisons of modeled and observed $SO_4^{2-}$ levels. Since the primary aim of this study is to assess the impact of degassing volcanic SO₂ emissions on $SO_4^{2-}$ and since the modeling performance in Table 2 showed generally good agreement compared to observations over U.S.A. and Asia, we do not feel that inclusion of model performance for $NO_3^-$ and $NH_4^+$ would add significantly to the primary goals of the study. We thus would like to keep our focus on $SO_4^{2-}$ concentrations throughout the manuscript.**

- The conversion from SO2 to sulfate should also be discussed, since their different deposition characters (dry deposition velocity, wet scavenging). I would suggest the authors focus on the total S instead of the sulfate only.

  **Reply:**

  **We agree with the reviewer that given differences in deposition sinks for gaseous and particulate sulfur, from an atmospheric budget perspective it is important to assess the total sulfur as well as the relative amounts of SO₂ and $SO_4^{2-}$. By including additional evaluation of surface SO₂ concentrations and SO₂ VCD as suggested by both reviewers, we believe we now provide a more complete picture of model**

performance of both $SO_2$ and $SO_4^{2-}$ as well as total-S. The CASTNET Florida site reports both $SO_2$ and $SO_4^{2-}$, and we have now included additional analysis of conversion rate from $SO_2$ to $SO_4^{2-}$ at this site. The result of temporal variation of conversion rate is additionally plotted in Fig. 6 (c). Based on this result, we have added the following discussion on Section 3.3.

"At this CASTNET site, EVE419 in Florida, the conversion rate from $SO_2$ to $SO_4^{2-}$ were further examined. We use the ratio $SO_4^{2-}/(SO_4^{2-} + SO_2)$ as an indicator of the S(IV) to S(VI) conversion rate with higher values indicating greater levels of oxidation of $SO_2$ to $SO_4^{2-}$. The temporal variation of this conversion rate is plotted in Fig. 6 (b). As expected, in response to seasonal variations in intensity of atmospheric chemistry and oxidant levels, the observed conversion rate was lower in the cool season and higher in warm season, and this general feature was captured by model. The mean of conversion rate through the year was 0.58 in the observation whereas original H-CMAQ was 0.42. By incorporating degassing volcanic $SO_2$ emissions, the mean of conversion rate increased to 0.43, but still underestimated the observed value. Since data from routine measurement networks are not designed to specifically characterize impacts of volcanic emissions on atmospheric sulfur budgets, these observations are unable to unambiguously quantify any modulation in S(IV) to S(VI) conversion rates due to the presence of volcanic emissions as also evidenced by the small change in the estimated conversion rate between the simulations with and without volcanic degassing emissions."

- P4, L10. The global SO2 emission is described here. What about the other species, such as NOx, NH3, etc? Since the conversion from SO2 to sulfate, dry and wet deposition as well as atmospheric transport should be controlled by the reactions with the other species.
  **Reply:**
  **As we stated in the original manuscript, a total of 105.8 Tg/yr of $SO_2$ emissions are emitted over the modeling domain. We have analyzed $NO_x$ and $NH_3$ emissions in the same manner, and results were 93.5 Tg/yr for $NO_x$ and 45.5 Tg/yr for $NH_3$**

emissions. The basis for the emission estimates for all species used in our calculations are detailed in the references provided in the manuscript and are thus not repeated here. As mentioned in our response to the second specific comment, we believe that it is better to focus on $SO_2$ emission and $SO_4^{2-}$ concentration within the scope of this study. We thus have not incorporated any revisions in the manuscript in response to this comment.